# Towards All-Non-Vacuum-Processed Photovoltaic Systems: A Water-Based Screen-Printed Cu(In,Ga)Se_2_ Photoabsorber with a 6.6% Efficiency

**DOI:** 10.3390/nano13131920

**Published:** 2023-06-23

**Authors:** Bruna F. Gonçalves, Viviana Sousa, José Virtuoso, Evgeny Modin, Oleg I. Lebedev, Gabriela Botelho, Sascha Sadewasser, Laura M. Salonen, Senentxu Lanceros-Méndez, Yury V. Kolen’ko

**Affiliations:** 1International Iberian Nanotechnology Laboratory, 4715-330 Braga, Portugal; bruna.goncalves@inl.int (B.F.G.); viviana.sousa@inl.int (V.S.); jose.virtuoso@inl.int (J.V.); sascha.sadewasser@inl.int (S.S.); laura.salonen@inl.int (L.M.S.); 2Center of Physics, University of Minho, 4710-057 Braga, Portugal; senentxu.lanceros@bcmaterials.net; 3Center of Chemistry, University of Minho, 4710-057 Braga, Portugal; gbotelho@quimica.uminho.pt; 4International Iberian Nanotechnology Laboratory, Universidad Politécnica de Madrid, 28040 Madrid, Spain; 5CIC nanoGUNE, 20018 Donostia-San Sebastian, Spain; e.modin@nanogune.eu; 6Laboratorie CRISMAT, UMR 6508, CNRS-ENSICAEN, 14050 Caen, France; oleg.lebedev@ensicaen.fr; 7BCMaterials, Basque Center for Materials, Applications and Nanostructures, UPV/EHU Science Park, 48940 Leioa, Spain; 8Ikerbasque, Basque Foundation for Science, 48009 Bilbao, Spain

**Keywords:** all-non-vacuum processing, Cu(In,Ga)Se_2_, photovoltaics, screen printing, sustainable inks

## Abstract

During the last few decades, major advances have been made in photovoltaic systems based on Cu(In,Ga)Se_2_ chalcopyrite. However, the most efficient photovoltaic cells are processed under high-energy-demanding vacuum conditions. To lower the costs and facilitate high-throughput production, printing/coating processes are proving to be effective solutions. This work combined printing, coating, and chemical bath deposition processes of photoabsorber, buffer, and transparent conductive layers for the development of solution-processed photovoltaic systems. Using a sustainable approach, all inks were formulated using water and ethanol as solvents. Screen printing of the photoabsorber on fluorine-doped tin-oxide-coated glass followed by selenization, chemical bath deposition of the cadmium sulfide buffer, and final sputtering of the intrinsic zinc oxide and aluminum-doped zinc oxide top conductive layers delivered a 6.6% maximum efficiency solar cell, a record for screen-printed Cu(In,Ga)Se_2_ solar cells. On the other hand, the all-non-vacuum-processed device with spray-coated intrinsic zinc-oxide- and tin-doped indium oxide top conductive layers delivered a 2.2% efficiency. The given approaches represent relevant steps towards the fabrication of sustainable and efficient Cu(In,Ga)Se_2_ solar cells.

## 1. Introduction

Chalcopyrite Cu(In,Ga)Se_2_ (CIGS) is one of the most promising photoabsorber materials due to its high absorption coefficient, excellent photoelectric properties, tunable band gap, and photodegradation stability [1]. Moreover, for Cu(In_1−x_,Ga_x_)Se_2_ systems, when x = 0, the material exhibits a bandgap of 1.0 eV, and when x = 1, a bandgap of 1.7 eV is obtained. Importantly, the maximum theoretical efficiency for CIGS PVs is obtained with a bandgap of ~1.34 eV, and top-efficiency PV cells are produced using a ratio of [Cu]/[In + Ga] between 0.8 and 0.9 and a [Ga]/[In + Ga] ratio of 0.3 [2].

Recently, photovoltaics (PVs) based on CIGS thin film technology have reached an efficiency of 23.4% [3] when fabricated using vacuum-based production processes. However, the use of printing/coating deposition processes instead can lower the costs and enable high throughput for industrial production. These processes demand a low energy input and employ sustainable procedures, which render them attractive for the fabrication of PVs.

To date, CIGS solar cells with efficiencies of up to 17.3% have been produced using electrodeposition [4] and printing/coating [5] processes for photoabsorber deposition. The faster, more practical, and, in some cases, easily scalable printing/coating processes, such as spin coating [5,6], blade coating [7,8], inkjet printing [9,10], spray coating [11,12], and screen printing [13,14], have been successfully employed in solar cell fabrication. Despite the high efficiencies obtained, ranging from 6.1% [13] to 17.3% [5], environmentally friendly photoabsorber ink formulation could improve these processes from a sustainability point of view. To this end, toxic solvents, such as hydrazine [5,6], hexanethiol [7], ethanolamine [10], and ethylenediamine [12], have been replaced with non-toxic water and ethanol [11,15,16,17], albeit with a concomitant reduction in device efficiencies.

On the other hand, printable transparent top conductive layers are also attracting increasing interest. However, only a few reports have emerged on the production of all-non-vacuum-processed CIGS solar cells. The spin coating of the photoabsorber followed by the chemical bath deposition (CBD) of the CdS layer and spin coating of the top conductive layers of intrinsic zinc oxide (i-ZnO), aluminum doped zinc oxide (AZO), and silver nanowires (AgNW) delivered solar cells with efficiencies ranging from 1.6% [18] to 7.7% [19]. However, sustainability has never been addressed from the point of view of solvent toxicity.

In this work, we report on two procedures for the development of environmentally friendly CIGS solar cells. A device based on a screen-printed photoabsorber layer, CBD of CdS, and vacuum-deposited top conductive layers presented a reliable performance with 6.6% efficiency for the champion cell. The all-non-vacuum-processed device featuring spray-coated i-ZnO and tin-doped indium oxide (ITO) top contacts yielded a 2.2% efficiency. The procedures employed herein contribute to the development of more sustainable CIGS solar cells.

## 2. Materials and Methods

### 2.1. Materials

All chemicals were used as received: potassium cyanide (KCN, ≥98.0%, Sigma-Aldrich, St. Louis, MO, USA), thiourea (CS(NH_2_)_2_, ≥99.0%, Sigma-Aldrich, St. Louis, MO, USA), cadmium acetate (Cd(ac)_2_, 99.995%, Sigma-Aldrich, St. Louis, MO, USA), aqueous ammonium hydroxide solution (28–30%, Acros Organics, Geel, Belgium), tin-doped indium oxide 20% suspension in water (ITO, 18 nm with In_2_O_3_/SnO_2_ 90:10%, ≥99.0%, Sigma-Aldrich, St. Louis, MO, USA), ethanol (≥99.8%, Honeywell, Charlotte, NC, USA), acetone (≥99.5%, Honeywell, Charlotte, NC, USA), and isopropanol (≥99.8%, Honeywell, Charlotte, NC, USA). Ultrapure water (MQ water, 18.2 MΩ cm) was generated using a Milli-Q Advantage A10 Millipore system (Millipore, Burlington, MA, USA).

### 2.2. Top Conductive Ink Formulation

For i-ZnO ink formulation, 0.85 g of ZnO particles synthesized using a reported method [20] were dispersed in 10 mL of ethanol using ultrasonication at room temperature (RT) for 6 h. For ITO ink formulation, 2.5 mL of the commercial dispersion in water was mixed with 2.5 mL of ethanol using vortex and ultrasonication at RT for 20 min.

### 2.3. Solar Cell Fabrication

The CIGS thin film deposition was performed using our previously described method [21]. Briefly, the photoabsorber thin film was deposited with a semi-automatic screen printer (DX-3050D, DSTAR) using a 180-thread cm^−1^ mesh count and 85-shore squeegee at a 0.3 m s^−1^ velocity and 75° deflection angle. Two-step printing was performed on conductive fluorine-doped tin-oxide (FTO)-coated soda lime glass (SLG) substrates (FTO/SLG) of 2.5 × 2.5 cm^2^ (7 Ω sq^−1^, Dyesol) that were previously cleaned sequentially with acetone, isopropanol, and water via ultrasonication (Elmasonic P30H) at 60 °C for 20 min each. The thin film was selenized at 550 °C using a heating ramp of 55 °C min^−1^ for 15 min in total inside a graphite box, together with Se shots using a tubular furnace flushed with 5%H_2_/Ar at 100 sccm. At the end of the selenization program, the furnace lid was completely opened for fast cool down. Finally, the prepared films were etched with 5% aqueous KCN solution at RT for 30 s, a commonly used procedure reported in the literature to guarantee the absence of a CuSe detrimental phase and a uniform deposition of CdS over the CIGS. Next, a CdS buffer layer of ~70 nm in thickness was deposited via CBD. For this purpose, a precursor solution containing 85 mL of ultrapure water, 15 mL of ammonium hydroxide, 0.9% Cd(ac)_2_ water solution, and 8% thiourea was heated at 60 °C in a water bath. The samples were placed inside the precursor solution for 7 min and moved up and down for the first 10 s of each minute. After finishing the procedure, the samples were rinsed with water and dried under a N_2_ flow. The i-ZnO window layer was sputtered using a rotating stage at 10 rpm with 20 sccm of Ar flow at 60 W for 50 min, providing a final layer thickness of 50 nm. The AZO transparent conductive layer was sputtered under an Ar flow of 20 sccm and 60 W for 42 min, providing a final thickness of 200 nm. The fabricated solar cell was finalized by scratching the edge of the cells with a scalpel down to the FTO back contact. Next, a thin layer of metallic indium was welded onto the scratched place to improve the electric contact between the cell and the probes used for the PV performance measurements.

The all-non-vacuum-processed solar cells were produced as described above, up to and including the CdS layer. The top conductive layers used for this cell were i-ZnO and ITO. The i-ZnO layer was deposited via one-step spray coating and dried at 120 °C for 1 h to improve the crystallinity. The ITO layer was spray-coated using two steps followed by drying at 100 °C to ensure solvent evaporation. The spray-coating process was performed manually using an airbrush gun (Dexter) powered by compressed air, vertically positioned 20 cm above a hotplate at 90 °C using a zig-zag coating direction. As a comparison for the spray-coated device, sputtering of the ITO was also performed using a rotating stage at 10 rpm with 20 sccm of Ar flow at 60 W for 4 h, providing a final layer thickness of 200 nm.

### 2.4. Characterization

#### 2.4.1. Electron Microscopy

A cross-sectional morphological evaluation of the resultant solar cells and the deposited top conductive layers was performed via scanning electron microscopy (SEM) using a Quanta 650 FEG ESEM and Helios NanoLab 450S DualBeam microscopes (FEI), fitted with spectrometers for energy-dispersive X-ray spectroscopy (EDX). The lamellae used for the cross-sectional evaluation of the solar cells were prepared using the focused ion beam (FIB) method. The evaluation of the chemical composition and fine microstructure of the solar cell layers was performed via high-angle annular dark-field scanning transmission electron microscopy (HAADF–STEM) and EDX in the STEM mode (STEM−EDX) using a JEM-ARM200F cold FEG probe and an image-aberration-corrected microscope (JEOL) equipped with a large-angle CENTURIO EDX detector and QUANTUM GIF operated at 200 kV.

#### 2.4.2. Optical Properties

The optical transmittance of the ITO transparent conductive layer was analyzed using a LAMBDA 950 UV–Vis–NIR spectrophotometer (PerkinElmer) equipped with a 60 mm integrating sphere and an InGaAs detector at room temperature.

#### 2.4.3. J-V Characterization

The current density–voltage (J–V) curves of the solar cells were measured with a four-point probe system Oriel Sol3A Class AAA Solar Simulator (Newport) with a 100 mW cm^−2^ illumination source using an AM 1.5 filter. For each sample, 10 cells with an area of ~0.16 cm^2^ each were isolated and measured using a two-probe system, placed on the top contact active area cell of AZO/ITO and on the indium-welded back contact. Prior to the measurement, the light intensity was calibrated using a mono-silicon reference cell with a BK7 optical window calibrated by the National Renewable Energy Laboratory (NREL). A 2.5 × 2.5 cm^2^ shadow mask was used to cover the active area in order to avoid interference from both scattering light and adjacent current leakage.

## 3. Results

To increase the sustainability of the CIGS PVs, environmentally friendly inks were formulated for both the photoabsorber and top conductive layers. To deposit the photoabsorber, we used our previously developed procedure [21] comprising the wet ball milling of Tiron-functionalized Cu, In, and Ga oxide nanoparticles in water. To this end, the dried and ground powder was dispersed in a solution of hydroxypropyl methyl cellulose (HPMC) and a mixture of green solvents of water/ethanol for ink formulation. Then, two-step screen printing of this ink onto a previously cleaned FTO/SLG substrate was carried out, followed by one-step selenization. During selenization, the growth of the CIGS is a complex process, where the binary selenides of Cu and In grow initially, followed by the sequential growth of the CuInSe_2_ and the CuGaSe_2_ phases. Finally, at temperatures above 500 °C, slow interdiffusion between the CuInSe_2_ and CuGaSe_2_ takes place to form the CIGS crystal, along with Ga segregation towards the back contact [2]. A compact thin film was obtained with a thickness of ~2.5 µm, consisting of pure-phase CIGS with a tetragonal structure, as detected via X-Ray diffraction (XRD) and Raman spectroscopy, without signs of chalcopyrite secondary phases (Figure 1a–c). However, as a consequence of the selenization procedure, a small admixture of SnSe_2_ deriving from the FTO back contact was detected in the Raman spectrum (Figure 1b). The resulting chemical composition obtained via SEM-EDX and the optical band gap were found to be Cu_0.92_(In_0.77_Ga_0.31_)Se_2_ (Figure 1d) and 1.08 eV, respectively, rendering the photoabsorber an excellent candidate for sustainable CIGS solar cell production [21].

The solar cell containing the screen-printed CIGS thin film was finished by first depositing a CdS layer through CBD to complete the p-n junction and to function as a buffer for the correct deposition of the upper layers. Then, an i-ZnO layer and, subsequently, a transparent conductive oxide layer of AZO were sputtered on top for the charge carrier collection. A cross-sectional analysis of the device with the stack SLG/FTO/CIGS/CdS/i-ZnO/AZO was performed using an FIB SEM specimen preparation (Appendix A), revealing a solar cell with a thickness of ~3 µm with well-stacked layers. The chemical composition was further analyzed via EDX mapping in the STEM mode (Figure 2). Starting from the top layer, a small intermixing of CdS with the CIGS photoabsorber was found, with the presence of Cd and S elements in some of the pores throughout the photoabsorber. Nevertheless, CdS was found to accomplish its function as a buffer layer, preventing the penetration of Zn and O elements from the top conductive layers to the photoabsorber. Notably, a uniform distribution of Cu, In, Ga, and Se was found throughout the photoabsorber, with no signs of the presence of unreacted metal oxides, suggesting a complete conversion of Cu, In, and Ga oxides into the CIGS crystal. Nevertheless, due to the small grain size of the particles, the photoabsorber layer was found to be porous.

In the chemical composition analysis of the device, a migration of Sn from the FTO back contact into the CIGS layer was also detected as the presence of voids at the FTO–CIGS interface, similar to our earlier findings [13]. This migration can be attributed to the unavoidable reaction of Sn with Se during selenization, as detected in the Raman data, which gives rise to the aforementioned voids at the interface. Compared with our previous work [13], a lower number of such defects were found in the present device, which we attribute to the elimination of the calcination step used in the previous report.

The PV performance of the device was measured using J–V curves on ten solar cells in total with an area of 0.16 cm^2^ each, under light and dark conditions (Appendix A), demonstrating that we produced a reliable and promising CIGS solar cell with an average efficiency of ~4.5 ± 0.9%. The champion solar cell exhibited a remarkable efficiency of 6.6% (Figure 3), a short-circuit current (JSC) of 36.7 mA cm^−2^, an open-circuit voltage (VOC) of 0.34 V, and a fill factor (FF) of 50.2%. The high JSC can be associated with the low band gap of the photoabsorber CIGS thin film of 1.08 eV. Similarly, the low VOC is related to recombination losses and the low bandgap. A similar PV behavior has been observed in the literature for CIGS photoabsorber layers prepared by employing a selenization procedure [15,22,23,24]. The achieved moderate FF can be attributed to losses stemming from series resistance.

Despite the promising results, the PV performance of the presented device could be further improved through additional optimization. Importantly, the aforementioned porous photoabsorber and the voids in the FTO–CIGS interface are potential regions of high recombination of charge carriers resulting in a low VOC, thus negatively impacting the PV performance.

To further increase the sustainability of the CIGS solar cells, we targeted the replacement of the vacuum-based processes used in the deposition of the top conductive layers of i-ZnO/AZO with non-vacuum-based ones. We opted for the use of spray coating due to its simplicity and the absence of thickener additives in the ink formulation. To this end, i-ZnO nanoparticles were synthesized following a reported procedure [20] and dispersed in ethanol using ultrasonication.

ITO is the most commonly used transparent conductive oxide (TCO) in the industry due to its convenient processability and good optical and electrical properties, both desirable characteristics for the cost-effective industrial production of PVs [25,26]. Thus, ITO was selected as the window layer for our CIGS solar cells. For the i-ZnO and ITO inks, one- and two-step coating, respectively, was found to be sufficient to achieve a full coverage of the layers below. The cross-sectional images of the spray-coated i-ZnO layer revealed a rough layer with a thickness of ~100 nm (Appendix A).

The top conductive layers of solar cells greatly impact the performance of the device, and in order to allow for an efficient absorption of photons by the p-n junction, these layers should feature high transparency in the visible region (>80%) and good electrical properties [27]. Using cross-sectional SEM imaging (Figure 4a), the ITO layer was found to be a compact film with a thickness of ~340 nm. The UV–Vis–NIR measurements revealed that the range of highest optical transmittance for the spray-coated ITO layer matched well with the range of the highest absorption of photons by the CIGS photoabsorber thin films (400–1100 nm) (Figure 4b). Moreover, the observed optical transmittance in the visible region (λ = 550 nm) was found to be around ≈70%, which is slightly lower than the reported optimal value (>80%). The obtained low transmittance probably stemmed from residuals left from the ITO solution.

The impact on performance when replacing the sputtered ITO with a spray-coated one was studied by measuring the J–V curves of the solar cells with the screen-printed photoabsorber, chemical-bath-deposited CdS, and spray-coated i-ZnO (Figure 5). We evaluated 20 solar cells in total under light and dark conditions (Appendix A). The highest PV performance was found for the cell with the sputtered ITO layer (Figure 5a), with a 5.6% efficiency for the champion device and an average efficiency of 4.7 ± 0.6%, which was attributed to the good deposition uniformity provided using this technique. The use of the spray-coated ITO layer (Figure 5b) resulted in an average efficiency of 1.6 ± 0.2% and a 2.2% efficiency for the champion device, with a JSC of 9.4 mA cm^−2^, VOC of 0.34 V, and FF of 67.5%. Cross-sectional SEM imaging showed the stack of an all-non-vacuum-processed CIGS solar cell with spray-coated i-ZnO and ITO layers with proper stacking of all the layers (Appendix A).

The sputtering of ITO gives rise to films with a higher conductivity than those obtained using non-vacuum deposition processes [28,29]. Thus, a lower series resistance is expected for the sputtered ITO layer than for the spray-coated one, which is reflected in the resultant J–V curves. On the other hand, the device with sputtered ITO revealed a lower shunt resistance due to manufacturing defects, which mainly resulted in a lower average FF obtained for the ten solar cells measured: 57.6% and 74.0% for the sputtering and spray-coated ITO, respectively. The excellent optical and electrical characteristics of the sputtered ITO layer gave rise to a higher transmittance of photons that could be absorbed by the CIGS layer and a higher extraction of charge carriers by the top conductive layer itself, explaining the higher JSC of 20.3 mA cm^−2^, compared with 9.4 mA cm^−2^ obtained using the spray-coated ITO. Previous studies have revealed that due to differences in ITO crystallinity, higher bandgaps are obtained when using sputtering as compared to non-vacuum processes [30]. Moreover, controlling the indium content is difficult with non-vacuum deposition, contributing to slightly different VOC values of 0.39 V and 0.34 V when using sputtering and spray coating, respectively.

## 4. Discussion

Due to the high potential of CIGS solar cells, they are under constant optimization on many different levels, from device architecture to materials engineering, including increasing attention devoted to non-vacuum-processed devices. Electrodeposition has been employed to produce solar cells with a 17.3% efficiency [4]. A similar efficiency was achieved using spin coating [5], while blade coating [7], inkjet printing [9], and spray coating [11] resulted in devices with 15.0%, 11.3%, and 10.7% efficiencies, respectively.

To date, few solar cells have been fabricated using screen printing, despite its great promise for the deposition of large-dimension layers with a low cost, good uniformity, and high resolution, characteristics that are highly attractive for industrial production. Furthermore, this technique allows one to print layers with a thickness of a few micrometers on any kind of substrate, from flexible polymers to rigid glasses. Solar cells with efficiencies ranging from 2.4% [14] to 6.1% [13] have been achieved with screen-printed CIGS photoabsorbers.

The few existing studies on all-non-vacuum-processed CIGS solar cells comprising solution-processed CIGS photoabsorbers have reported efficiencies of 7.7% [19] and 1.6% [18]. The former was produced by spin coating the FTO back contact on glass substrate [19]. A CIGS precursor layer was then spin-coated and selenized, albeit without specifying the nature of the precursors or the ink formulation. To finalize the cell, CBD of the CdS and spin coating of the i-ZnO/AgNW layers were performed. The lower-efficiency CIGS solar cell was produced by spin coating all the layers [18]. First, a CIGS precursor ink comprising synthesized CIGS nanoparticles suspended in o-dichlorobenzene was deposited on commercial molybdenum-coated glass followed by calcination to remove the organic matter, without the need for a selenization step. Then, the CdS, ZnO, AgNW, and ZnO layers were spin-coated consecutively, followed by final annealing at 200 °C. The ZnO precursor solution was developed using 2-methoxyethanol and monoethanolamine solvents.

Our work presents the combination of a screen-printed water-based CIGS photoabsorber with chemical-bath-deposited n-type CdS and sputtered i-ZnO/AZO layers, leading to reliable CIGS solar cells with an average efficiency of ~4.5 ± 0.9% and a maximum efficiency of 6.6%. However, the porous CIGS layer and the detected voids at the interface between the back contact and the photoabsorber negatively impact the performance. To eliminate these potential recombination spots, an adapted selenization procedure (temperature, time, and atmosphere) or additional sulfurization may increase the grain size and, therefore, the results for a densely packed photoabsorber layer, increasing the VOC and, concomitantly, the PV performance [31]. Regarding the voids at the FTO–CIGS interface, the back contact–photoabsorber interface should be improved. Notably, we previously showed [21] that among molybdenum-coated glass, bare stainless steel, carbon-coated stainless steel, bare graphite, graphene-coated graphite, and FTO, the latter is the most suitable back contact for use with the deposition methodologies presented herein. Possibly, the implementation of a passivation layer on top of the FTO back contact may provide a solution to prevent SnSe_2_ formation and the resulting voids, thus improving the PV performance of the device. Nevertheless, to the best of our knowledge, the presented device sets the record efficiency for CIGS solar cells with screen-printed photoabsorbers. Interestingly, our previously reported screen-printed CIGS solar cells [13] were prepared using an ink formulation based on terpineol, and the process included a high-energy-demanding calcination step to eliminate the carbon residues from the thickeners, as well as a selenization step to grow the crystal. In contrast, this work used water and ethanol as solvents, thus omitting the calcination step prior to selenization, since the carbon residues were eliminated during the selenization.

Recently, in addition to the coating/printing of the CIGS photoabsorber layer, there has been increasing interest in non-vacuum-processed upper layers to provide access to all-non-vacuum-processed CIGS solar cells [18,19]. This pathway results in CIGS PVs with lower costs and allows for high-throughput production. The CBD of the CdS layer is a highly well-established non-vacuum-based process used in top-efficiency CIGS solar cells. Furthermore, the CBD of CdS has been demonstrated to deliver layers with a good photoconductivity and coverage [32]. Nevertheless, the CBD of Zn(O,S) as a non-toxic alternative buffer layer is already well-established and has led to a record device performance on vacuum-processed CIGS [3].

On the other hand, the TCOs used as top conductive layers are under extensive investigation due to their utility in devices of different natures, such as displays and smart windows [33]. The upper layers of vacuum-processed CIGS solar cells have been replaced with spray- and spin-coated carbon allotropes [34], AgNW [35], ITO nanoparticles [36], PEDOT:PSS [37], and AZO [25], all in combination with i-ZnO or similar semiconductors. Herein, we replaced the sputtering of i-ZnO/AZO layers with spray-coated i-ZnO/ITO layers, leading to a 2.2% efficient device. In line with our sustainable approach, both inks were developed using ethanol and water as solvents. When replacing the spray-coated ITO with a sputtered ITO layer, the CIGS solar cells showed an efficiency of 5.6%, indicating that the spray-coated ITO strongly impacts performance and requires further optimization.

## 5. Conclusions

This work presented a robust and reliable method that can be used to fabricate environmentally friendly CIGS solar cells including a screen-printed photoabsorber. The used ink formulation is based on ball-milled Cu, In, and Ga oxides dispersed in water/ethanol solvents and a nature-derived thickener (HPMC). After the screen printing of the oxide layer, one-step selenization was employed to form a pure-phase CIGS crystal working as a photoabsorber layer with a [Cu]/[In + Ga] ratio of 0.92 and a [Ga]/[In + Ga] ratio of 0.31, ratios typically found for top-efficiency devices. Above the CIGS layer, a buffer n-type layer of CdS was deposited using chemical bath deposition to complete the p-n junction. Finally, i-ZnO and AZO layers working as window layers were deposited through sputtering. The developed device, with a final stack comprising SLG/FTO/CIGS/CdS/i-ZnO/AZO, sets a new record for printed CIGS solar cells with a maximum efficiency of 6.6%, with the top conductive layers deposited through vacuum-based sputtering.

To further minimize the environmental impact of the solar cells’ production, we also deposited the top conductive layers through the spray coating of water-/ethanol-formulated i-ZnO and ITO inks. The obtained CIGS solar cell featured an efficiency of 2.2%, the highest reported using environmentally friendly all-non-vacuum processes. This work supports the use of a combination of printing/coating technologies with sustainable inks to lower production costs, allowing for high throughput and, in general, providing access to more sustainable CIGS solar cells.

## Figures and Tables

**Figure 1 nanomaterials-13-01920-f001:**
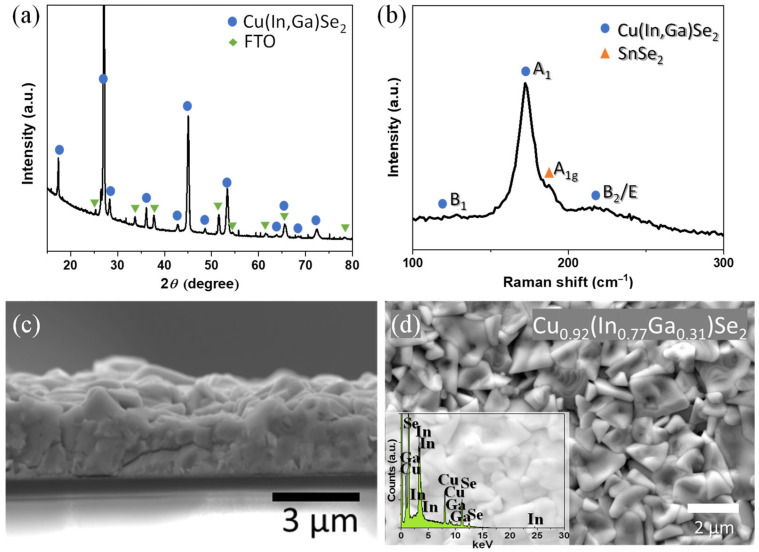
Structural and compositional characterization of the resultant CIGS photoabsorber layer deposited onto FTO/SLG substrate using HPMC oxide ink: XRD pattern (CIGS—International Centre for Diffraction Data (ICDD), card no. 01-082-9226, tetragonal, I-42d; SnO_2_ from FTO—ICDD card no. 04-003-5853, tetragonal, P42/mnm); (**a**), Raman spectrum; (**b**), cross-sectional SEM image; and (**c**) surface SEM image with the corresponding metal ratio (**d**) [21].

**Figure 2 nanomaterials-13-01920-f002:**
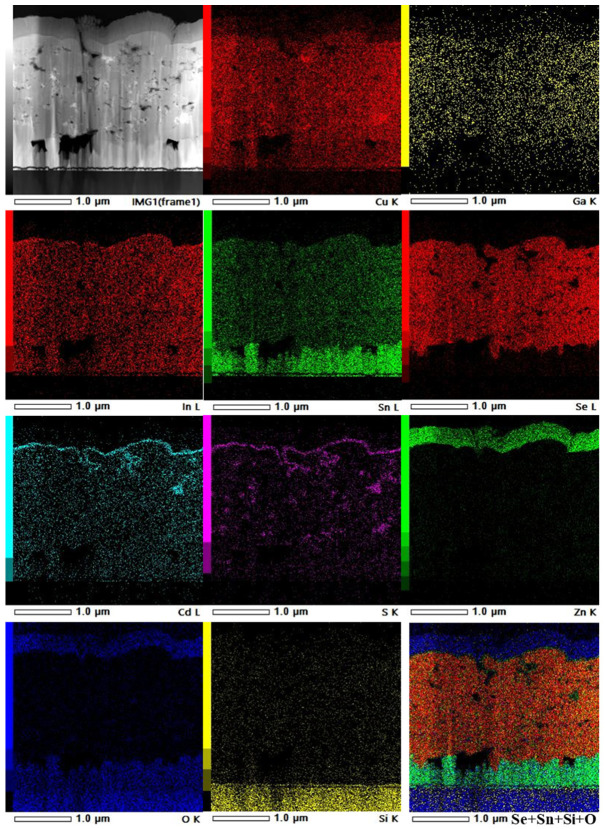
Cross-sectional HAADF-STEM image of the champion solar cell: SLG/FTO/CIGS/CdS/i-ZnO/AZO, with the collected EDX maps of Cu K, Ga K, In L, Sn L, Se L, Cd L, S K, Zn K, O K, and Si K elements and Se+Sn+Si+O mixture color image.

**Figure 3 nanomaterials-13-01920-f003:**
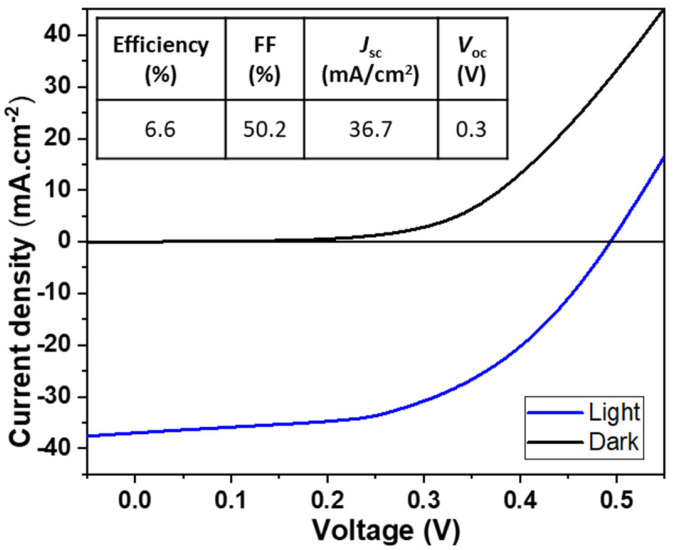
The light and dark J−V curves of the champion CIGS solar cell.

**Figure 4 nanomaterials-13-01920-f004:**
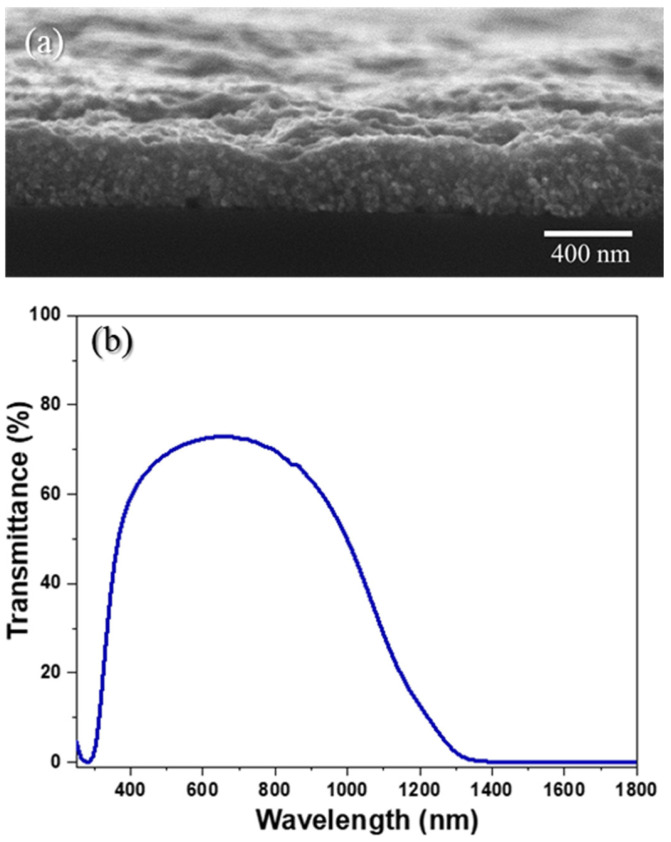
Cross-sectional SEM imaging of the ITO layer coated on SLG (**a**), together with the respective transmittance UV–Vis–NIR spectra (**b**).

**Figure 5 nanomaterials-13-01920-f005:**
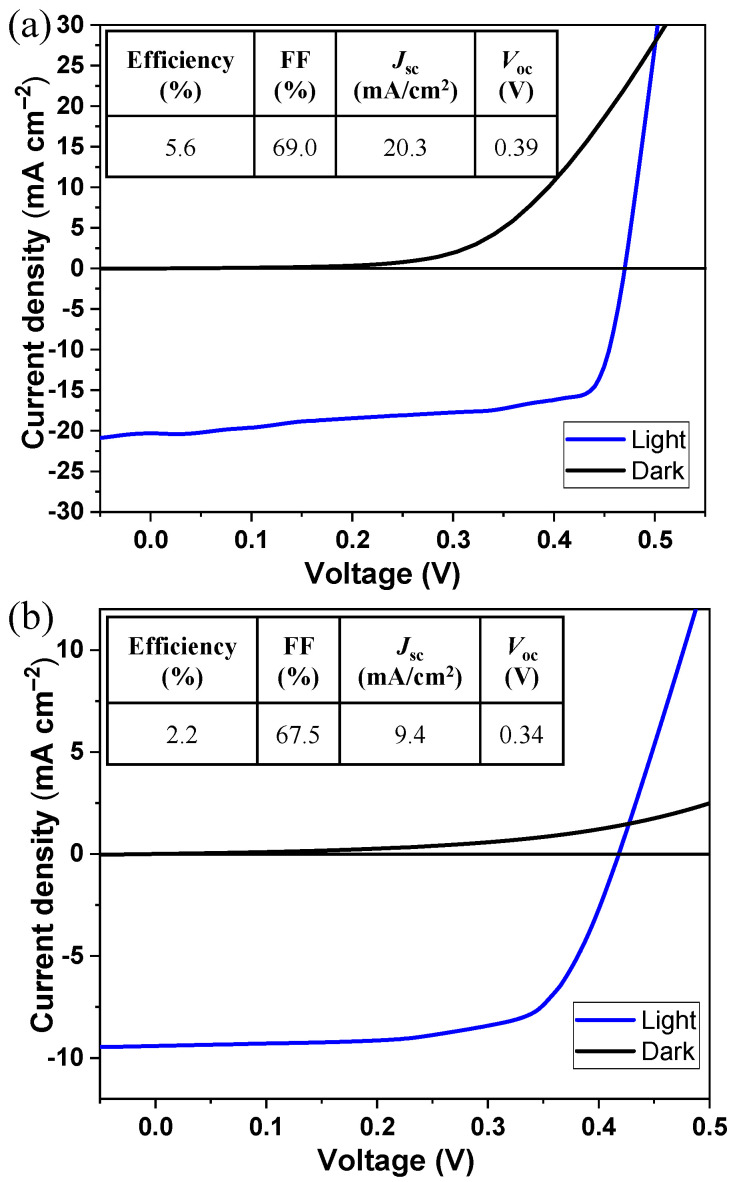
The J−V curves of champion CIGS solar cells developed with sputtered ITO (**a**) and spray-coated ITO (**b**).

## Data Availability

No new data were created or analyzed in this study. Data sharing is not applicable to this article.

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
