# Peer review of "Towards All-Non-Vacuum-Processed Photovoltaic Systems: A Water-Based Screen-Printed Cu(In,Ga)Se2 Photoabsorber with a 6.6% Efficiency"

_nanomaterials, 2023, doi:10.3390/nano13131920_

Round 1
Reviewer 1 Report
The authors present a paper on the realization of CIGSe-based thin film solar cell produced in different ways, in particular the deposition of the window layers by means vacuum and non vacuum techniques.
A maximum efficiency of 6.6% is obtained by a double step of screen-printed CIGSe ball-milled powders dispersion (in HPMC and water/ethanol solution) closed with CdS via CBD and sputtered i-ZnO and Al:ZnO while the value of 2.2% is reached by the spray coating of i-ZnO and ITO. As comparison, an ITO layer was deposited by sputtering onto another spray-coated i-ZnO layer on the same CIGSe-based structure with the best PCE value of 5.6%.
The work is clear and well written and can be considered for publication after some minor revisions:
- control superscript and subscript;
- check language and typos.
Just a question about the buffer layer: since your main purpose is to make sustainable and green the production method of CIGSe-based solar cells, why did not test a Cd-free buffer layer? Are you studying this?
Thank you and good luck for your next works
Minor revisions for the language and typos are requested, for example:
- line 50 page 2: "...end..." could be changed in "...aim..."
- line 99,102,104,117.... page 3: "...during..." should be "...for..."
Author Response
The revisions are attached

Reviewer 2 Report

I appreciated this work. In conclusion, the authors should improve this paper.
Author Response
The revisions are attached in the next word document.
